# Biological Durability of Acetylated Hornbeam Wood with Soil Contact in Hungary

Fanni Fodor [1,*], Miklós Bak [1], András Bidló [2], Bernadett Bolodár-Varga [2] and Róbert Németh [1]

1 Institute of Wood Technology and Technical Sciences, Faculty of Wood Engineering and Creative Industries, University of Sopron, Bajcsy-Zsilinszky St. 4, H-9400 Sopron, Hungary; bak.miklos@uni-sopron.hu (M.B.); nemeth.robert@uni-sopron.hu (R.N.)

2 Institute of Environmental Protection and Nature Conservation, Faculty of Forestry, University of Sopron, Bajcsy-Zsilinszky St. 4, H-9400 Sopron, Hungary; bidlo.andras@uni-sopron.hu (A.B.); varga.bernadett@uni-sopron.hu (B.B.-V.)

* Correspondence: fodor.fanni@uni-sopron.hu

**Abstract:** This test aimed to discover if industrially acetylated hornbeam can tolerate real-field conditions in Hungary, where various microorganisms can attack the wood separately or cooperatively. Untreated samples accompanied the modified wood to assess the degradation capacity of the soil. The test also focused on weather parameters, the Scheffer index, and soil properties. The untreated stakes showed insect damage, soft rot decay, white rot decay, wasp stripping, moss, and cracks. All of the beech and hornbeam stakes broke after 3.5 years, and the last Scots pine sapwood stakes broke after 6 years. To date, acetylated hornbeam exhibits stronger resistance than untreated hornbeam, beech, and Scots pine sapwood. The acetylated hornbeam stakes showed no decay after 6 years of exposure, and they became dry shortly after being taken from the soil. Acetylated hornbeam stake number 7 had superficial brown rot decay after 18 months, which gradually worsened over the years. The Fourier transform infrared spectroscopy analysis revealed that this stake had lower acetyl content. It was associated with hornbeam wood; it had a wet pocket or a part that was not as permeable and achieved a lower grade of acetylation.

**Keywords:** wood acetylation; hornbeam; weather; soil; durability; microscopy

## 1. Introduction

The durability of natural and treated wood can be examined by laboratory tests (EN 113: 1996) and field tests (EN 330: 1993, ENV 807: 2001, CEN/TS 12037: 2003, AWPA E 18: 2004, EN 252: 2012), which allows it to be classified in durability classes (CEN/TS 15083: 2005, EN 350: 2016) and use classes (ISO 21887: 2007, EN 335: 2013).

In this research, a field test, specifically an in-ground test was used, where the wood was partly in contact with the soil, and the impact of atmospheric and biological factors was direct. Non-destructive methods such as visual assessment, macroscopic evaluation, density and mass loss, sound (sonic tomography), optical methods (X-ray tomography), microscopy, radiography, or destructive methods such as the pick/splinter test or strength tests can all assess wood degradation [1,2].

Visual assessment includes the determination of discolorations, cracks, mycelium, and insect damage. Microscopic detection is also possible if the mass loss is at least 5–10%. The signs of decay by soft rot, brown rot, white rot and tunneling bacteria can be well distinguished by their characteristic decay patterns [1,3].

Density is closely correlated with strength properties; thus, it can be a rough decay indicator during decay testing [1]. However, the density loss caused by white and brown rot is not comparable because white rot causes a small change in volume while brown rot causes substantial volume reduction. Determining mass loss is also a common method, but

it cannot detect the early stages of decay. For this, measuring the strength toughness and impact bending strength are the most sensitive methods [1].

During a pick-test, a pointed knife is poked into the specimens and withdrawn. The fracture characteristics of the splinter and the depth and appearance are assessed visually. Decay types are identified and evaluated according to EN 252: 2015. Failed specimens are analyzed microscopically to identify the type of decay [4].

The degradation rate caused by microorganisms depends on many factors, such as wood structure, natural or artificial toxicity of the wood, tolerance to temperature, moisture content, pH, and oxygen range. Although fungi and bacteria can degrade the wood, *Basidiomycetes* are usually more aggressive but less tolerant of extreme conditions than the bacteria or soft rot fungi. When colonizing wood, bacteria, molds, blue stain, and soft rot fungi decay initially, and then *Basidiomycetous* fungi take over [1].

Several authors have reported that acetylation of lignocellulosic materials increases their resistance to biological degradation. Laboratory tests and field tests have shown considerably improved protection against attack by white rot, brown rot, soft rot fungi, tunneling bacteria, and marine borers in various species such as southern yellow pine, beech, Scots pine, and poplar [5–12]. For example, beech, which has similar structure to that of hornbeam, showed no signs of decay at WPG 17% [11].

The acetylation rate can be expressed as the weight percentage gain (WPG) or the acetyl content, which is usually determined by high-performance liquid chromatography (HPLC) or near-infrared (NIR) analytical methods. The former is used in laboratory tests, while the latter is used on an industrial level. The higher the WPG, the higher the durability [11,13]. Studies have reported that the resistance of acetylated wood (with a WPG of ca. 20%) to fungal decay was equivalent to chromated copper arsenate (CCA)-treated wood at a high retention level (e.g., 103 kg/m$^3$) after 18 years in soil contact [14,15]. To provide sufficient durability, a WPG of at least ~10% needs to be reached for soft rot and white rot and ~20% for brown rot and tunneling bacteria [10–12].

Fungal colonization requires moisture for oxalate production. Moisture is also necessary for movement in the cell wall, in Fenton chemistry to degrade cell wall polysaccharides, for gene expression and enzyme activity, for glycoside hydrolysis, and for the movement of soluble nutrients. Wood-rotting fungi and insects have their specific enzyme systems, which degrade wood polymers into digestible units. Hydroxyl groups are biological enzymatic reaction sites; if these are chemically changed, the fungal enzymatic action cannot take place [13,16,17].

Among the chemical components, hemicellulose polymers have the highest moisture content (38%) compared to lignin (16%) and cellulose (12%) at 90% relative humidity and room temperature. It seems that fungi access the moisture first in hemicelluloses, which may be the first to be degraded. The hemicellulose polymers are interconnected throughout the cell wall and can act as a hemicellulose moisture pipeline [11,18,19].

Since 2008, Accsys Technologies (Arnhem, The Netherlands) has been commercially acetylating radiata pine and selling it on the market as Accoya® wood. The biological durability of Accoya® wood is DC 1, the highest class according to EN 350: 2016. It exhibits considerably increased biological resistance against brown rot, white rot, and soft rot fungi [10–12,20,21]. It has a WPG of ca. 20%, and its fiber saturation point is below 15% [17,20]. It is very resistant to subterranean and Formosan termites [22] and marine borers [23,24]. On the other hand, it is not sold as marine/borer resistant wood [23]. In a 10-year-long ground stake test in Greece (southern Mediterranean zone), Accoya® wood exhibited very good performance with no visual signs of decay [25]. Its structural integrity was maintained to some extent, but the modulus of elasticity (MOE) and modulus of rupture (MOR) properties decreased considerably, by 33 and 30%, respectively. It was concluded that although there are no visible signs of decay, there could be a significant loss of mechanical properties in acetylated wood.

Acetylation could improve the durability and dimensional stability of hornbeam (*Carpinus betulus* L.) wood, which is mainly used as firewood due to its low quality. It can

be found all over Europe, except in the Mediterranean. It is a diffuse-porous wood species; it does not form colored heartwood and has low natural durability (DC 5 according to EN 350: 2016). Hence, it is not recommended for outdoor use [26,27]. Hornbeam also tends to spalt, which is a major consideration for lumber production. During weathering, it becomes grey in a short time, and surface checks become apparent. By improving its properties with acetylation, hornbeam could be used to expand the range of durable species for outdoor applications in Hungary such as oak, black locust, Scots pine, and larch [26].

Only a few research papers have focused on the acetylation of hornbeam to increase its durability. In a previous laboratory test [28], hornbeam and acetylated hornbeam were exposed to three different fungal cultures according to EN 113: 1996. Hornbeam had DC 4 against *Rhodonia (Poria) placenta*, DC 4–5 against *Coniophora puteana*, and DC 5 against *Trametes (Coriolus) versicolor*, while acetylated hornbeam's weight loss was below 1%, which makes it DC 1 against all three fungi according to EN 350: 2016. Another research study tested the durability of hornbeam acetylated with acetic acid and liquid formalin [29]. The study found that the most effective treatment to achieve DC 2–3 was 10% liquid formalin and 5% acetic acid against *Trametes versicolor*, *Coniophora puteana*, and *Chaetomium globosum*. In spite of this, microscopic studies revealed a scarce appearance of hyphae, so fungi were able to colonize in lumina of acetylated hornbeam [30].

This research aimed to see if acetylated hornbeam could be utilized as an outdoor product in real-field conditions, with many different microorganisms that can attack the wood separately or cooperatively. The test was conducted for 6 years according to EN 252, and it also considered soil and weather characteristics of the field, as well. The difference in the rate of degradation was determined visually, by microscopy, and by calculating density and mass loss.

## 2. Materials and Methods

### 2.1. Sample Preparation

The long-term field test was performed according to EN 252: 2015 with some slight modifications: the sample dimensions were changed from 25 × 50 × 500 mm to 20 × 50 × 300 mm (thickness × width × length) because the dimensions of raw material were limited. There were 12 stakes of each type: untreated hornbeam and industrially acetylated hornbeam [28], supplemented with beech and Scots pine sapwood according to standard. Acetylation was carried out at Accsys Technologies (Arnhem, The Netherlands) on 28 × 160 × 2500 mm boards [28]. The stakes were cut from these boards, having WPG levels ranging from 13.55 to 16.15%. The average WPG was 15.10 ± 1.03%. The beech and pine stakes indicated the intensity of the decaying mechanism of the soil. Table 1 shows the important characteristics and parameters of the stakes. The stakes were conditioned at 20 ± 2 °C and 65 ± 5% relative humidity before measuring their parameters and weight.

**Table 1.** Information about the stakes.

| Average | Acetylated Hornbeam | | Hornbeam | | Beech | | Scots Pine Sapwood | |
|---|---|---|---|---|---|---|---|---|
| | On 10 mm Surface | On 2 cm Surface | On 10 mm Surface | On 2 cm Surface | On 10 mm Surface | On 2 cm Surface | On 10 mm Surface | On 2 cm Surface |
| No. of annual rings | 5 | 10 | 4 | 8 | 4 | 8 | 5 | 9 |
| Air-dry density (kg/m$^3$) | 794 ± 49 | | 745 ± 45 | | 719 ± 7 | | 525 ± 19 | |
| Mean density(kg/m$^3$) | 804 ± 50 | | 822 ± 55 | | 799 ± 10 | | 614 ± 23 | |
| Moisture content (%) | 3.35 ± 0.06 | | 14.16 ± 0.96 | | 13.50 ± 0.16 | | 13.34 ± 0.50 | |

The stakes were buried in the outdoor exposure testing field at the University of Sopron (47°40′41.4″ N 16°34′32.6″ E) in April 2016 (Figure 1). The stakes were buried

half of their length, one by one from each type. The distance between stakes was 30 cm. The vegetation on the field was cut regularly, and no chemicals or herbicides were used during the test. The presence of wood-decaying fungi and insects was also observed during the evaluation.

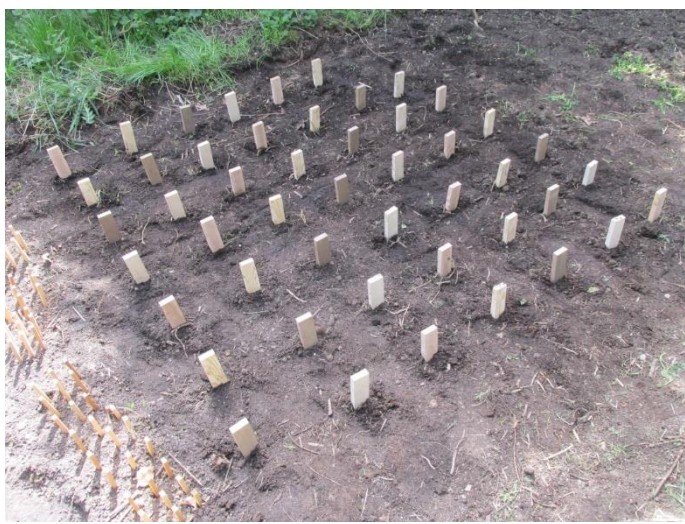

**Figure 1.** Stakes placed in soil in April 2016.

## 2.2. Soil of Testing Field

Five samples were taken from the soil of the testing field in order to examine the soil properties that influence the intensity of degradation. The samples were taken from four corners and the middle of the field.

Skeletal grain content was determined by their dry weight relative to the dry weight of the soil. These particles had low water-holding capacity compared to fine-sized fractions.

The measurement of soil pH was performed electrometrically at a soil/liquid ratio of 1/2.5 according to MSZ-08-0206-2: 1978. Then, 10 g of soil samples were soaked in distilled water and one molar solution of KCl, respectively. Soils with pH above 6 provide an ideal chemical environment for plant and fungal growth.

The calcium carbonate ($CaCO_3$) content of the soil was determined with a Scheibler calcimeter according to MSZ-08-0206-2: 1978.

Humus or organic matter content was measured by the wet incineration process using potassium dichromate for the oxidation of organic matter. Water and phosphorus iron were added. It was then back-titrated with 0.2 M Mohr's salt in the presence of ferric ion and diphenylamine sulfonic acid indicator. The organic carbon content of the soil was calculated based on the amount of Mohr's salt used in the titration. Microbial activity in soil is related to carbon stabilization and storage, water holding capacity, and aeration [31].

Particle content or fine-earth fraction was determined by the pipetting method according to MSZ-08-0205: 1978. Particles of fine earth fraction have a larger surface area and charge characteristics that promote soil aggregation, structural development, and the regulation of soil water movement [32].

## 2.3. Weather and Climate

The weather parameters of the testing field were received from the Department of Ecology and Bioclimatology of the University of Sopron. These included average and maximum monthly temperature, monthly precipitation, number of days with precipitation above 0.25 mm, sunshine duration per month, solar irradiance per month, and monthly relative humidity.

The Scheffer climate index (SCI) was also calculated, which Scheffer [33] proposed to estimate decay hazard by geographic location within the conterminous United States for wood exposed above ground to exterior conditions. The index is calculated from local

weather data using the mean monthly temperature and mean number of days with at least 0.25 mm of precipitation over the exposure period. An index of less than 35 represents the least favorable conditions for decay; 30 to 65, intermediately favorable conditions; and greater than 65, conditions most conducive to decay. As a metric by which relative hazard can be compared between geographic locations, the Scheffer index is not intended to predict the decay propagation rate or time to failure in specific constructions [34]. The rate of decay also depends on application, design, material resistance, soil water holding capacity, etc. This particularity is not necessarily a limitation of the approach, but it has been debated. However, the SCI is still, beyond doubt, the most frequently used index of its kind for estimating the relative climate-induced decay hazard of geographical locations [35].

### 2.4. Rate of Degradation

Every six months, the level of degradation of each stake was determined from 0 to 4 according to EN 252: 2015. Photos were taken before, during, and after exposure to observe the changes perceivable to the eye. After the failure of a stake, the broken pieces were dried at 103 °C in a drying kiln until a constant mass was reached, and the dry weight and parameters were determined to calculate the dry mass loss and dry density loss caused by degradation.

### 2.5. Microscopic Evaluation

Cubes with $10 \times 10 \times 10$ mm dimensions were cut from the lower part of the samples and dried at 103 °C in a drying kiln to constant weight. Then, they were placed into a desiccator. As the cubes were cut with a circular saw, which resulted in rough surfaces, the surfaces were smoothed with a razor blade/scalpel before examination with a Hitachi S-3400N PC-Based Variable Pressure Scanning Electron Microscope (Hitachi, Tokyo, Japan) and its software. Cross and longitudinal sections were examined as well. Microscopic analysis was performed at a 60 Pa vacuum and a 10 kV accelerating voltage using a BSE detector. The working distance was 10 mm. The surfaces were not coated with a sputter coater before imaging.

### 2.6. Fourier Transform Infrared Spectroscopy (FTIR)

Acetylated hornbeam stake number 7 showed local signs of decay (see Results and discussion), which was examined with Fourier transform infrared spectroscopy. The measurement was performed using a Specac Golden Gate ATR (attenuated total reflection) with zinc selenide (ZnSe) lenses. Sound (control) and decayed parts of the stake were ground separately into smaller pieces using mortar and pestle to improve the contact area of the sample with the diamond on the ATR FT-IR.

The FT-IR spectra were measured in transmittance mode (%T). A three-point baseline correction was applied to 850, 1180 and 1800 cm$^{-1}$ to maximum transmittance (100%). The peak intensity was determined at peaks 1740 cm$^{-1}$ and 1230 cm$^{-1}$. Then, it was translated to absorbance mode (%A), using the following formula: %A = $2 - \log$ (%T). Normalization was performed at peak 1030 cm$^{-1}$ having 1% absorbance. Then, the peak height ratios (PHR) were calculated by dividing the absorbance at 1740 cm$^{-1}$ and 1030 cm$^{-1}$ (%A$_{1740}$/%A$_{1030}$), and the absorbance at 1230 cm$^{-1}$ and 1030 cm$^{-1}$ (%A$_{1230}$/%A$_{1030}$). Both measurements for the sound part and decayed part were performed in triplicates. The rest of the stakes remained in the testing field for further studies.

## 3. Results and Discussion

### 3.1. Soil of Testing Field

Table 2 presents the summary of results regarding the testing field soil properties.

The skeletal grain content ranged from 4 to 11%, having an average of $7 \pm 3\%$. This does not significantly influence the water and nutrient holding capacity of the soil.

In the case of pH$_{H2O}$ or current acidity, there was no significant difference between the samples, as the values ranged from 7.1 to 7.3. This is neutral (<7.2) and slightly alkalic

(>7.2). As expected, $pH_{KCl}$ was lower than $pH_{H2O}$, and it ranged from 6.7 to 6.9. There was no significant difference between $pH_{KCl}$ and $pH_{H2O}$, which means there was no latent (hidden) acidity rate. From an agricultural point of view, the pH of this soil is sufficient for most plant species, as most nutrients are best absorbed by plants and are most mobile in this pH range.

**Table 2.** Soil characteristics of testing field. Average values of 5 samples are presented with variation in brackets.

| Skeletal Grain Content > 2 mm (%) | pH | | Calcium Carbonate (CaCO₃) Content (%) | Particle Content ≤ 2 mm | | | | Organic Matter Content (Humus) (%) |
|---|---|---|---|---|---|---|---|---|
| | H2O | KCl | | Clay (%) | Silt (%) | Fine Sand (%) | Coarse Sand (%) | |
| 7 | 7.2 | 6.8 | 3 | 19 | 18 | 43 | 20 | 4.0 |
| (3) | (0.1) | (0.1) | (0) | (1) | (4) | (1) | (4) | (0.8) |

The calcium carbonate ($CaCO_3$) content ranged from 2 to 3%, which corresponded to the expected values based on soil pH. This amount of $CaCO_3$ is favorable because it improves the soil structure. It is also advantageous in that there was no expected calcium deficiency.

Based on the tests, the amount of sludge (clay and silt fraction) ranged from 31 to 43%, with an average of 37%, which indicates a sandy loam type. This type has favorable water management properties because it allows water to enter well, retains it well, and makes it available for plants.

Humus or organic matter content was between 2.7 and 4.9%, having an average of $4.0 \pm 0.8$%, which is classified as a good/medium supply. This result also corresponded to the other soil properties.

According to these results, the soil in which the wooden specimens were tested was rich in nutrients and had good aeration and drainage properties. These characteristics were favorable for not just plant growth but also fungal growth, such as soft, white and brown rot fungi.

*Basidiomycetes* live in conditions with high oxygen content (soil with good aeration), moist wood with moisture content between 40 and 80%, and their optimal temperature range is from 24 °C to 32 °C. White rot fungi require higher moisture content and higher pH than brown rot fungi. They can decay wood during a short period when the temperature and the moisture are at optimal levels, e.g., summer, end of spring, and early autumn seasons. Soft rot fungi activate in a wide range of moisture contents, from relatively dry wood to saturated conditions, and a wide range of temperatures from 0 °C to 60 °C. They are active at a pH close to neutral and have better adaptability properties than other fungi during the whole year when soil moisture and temperature change periodically [1,12].

*3.2. Weather and Climate*

According to its solar-climatic classification, Hungary is situated about halfway between the Equator and the North Pole, in the temperate climatic zone. Hungary has a continental climate, with hot summers with low overall humidity levels but frequent showers and frigid to cold snowy winters. According to Péczely [36], Sopron is in the moderately cool–moderately dry climatic region.

Tables S1 and S2 list the weather parameters of the study site. The area of the study site has a warm and wet summer season (May–September) with mean temperatures between 13–23 °C, with a maximum of 37 °C, and monthly average precipitation between 63–85 mm. It has a drier winter season (October–April) with 25–54 mm of average precipitation per month, mean temperatures between −4–16 °C, with a maximum of 28 °C. The average annual rainfall during the test period was approximately 594 mm; the average annual temperature was 12 °C; the maximum temperature was 37.2 °C. The relative humidity ranged from 45 to 91%.

According to the ombrothermic diagram (Figure 2), there was no dry season during the year, which would be the area below the temperature line and above the precipitation line. On the other hand, the wet season—the area below the precipitation line and above the temperature line—was typical for the whole year.

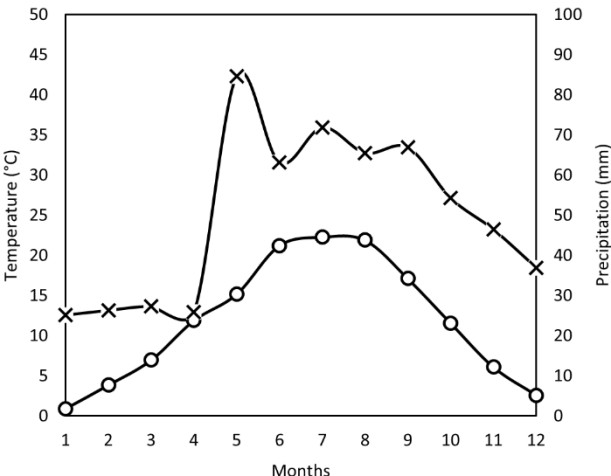

**Figure 2.** Ombrothermic diagram, which summarizes trends in temperature (O) and precipitation (X) of 6 years of exposure (2016–2022). The wet period is typical for the whole year; there is no dry period.

The wet and warm periods at the exposure site were quite long, which enabled fungal growth. For *Basidiomycetes*, the summer period was favorable, while for soft rot, the conditions for growth were good throughout the whole year when temperatures did not drop below zero. The moisture content of the soil and that of the wood specimens are related to each other, and the change in their moisture content is related to the oxygen level of the soil. The oxygen level and aeration properties of the soil enable the growth of microorganisms in the wood.

The Scheffer climate index of this site is 46.1, which indicates intermediately favorable conditions for decay. This index is similar to other reports regarding the SCI of different countries, including Hungary. The SCI of Hungary ranged from 35 to 50 in other investigations [37–39]. Compared to European countries, it is in about the middle range. Those territories that have the highest decay potential (so-called hot spots) are on the west coast of Norway, Ireland, the UK, and France, reaching a maximum SCI of 81 [38]. These territories will expand in the future, as the activity of wood degrading organisms (e.g., termites) is expected to increase in Europe as a result of warmer climate [39].

The decay rate of this testing field can also be perceived by the results of other field tests in the past. In a 6-month-long test [40], Turkey oak, hornbeam, Pannonia poplar, and Scots pine stakes with dimensions of 20 mm × 20 mm × 300 mm were tested. These stakes were heat-treated for 5 hours at 200 °C. After 6 months of exposure, the decay rating decreased after heat treatment from 1.6 to 0.9 for Scots pine, 1.4 to 0.3 for Turkey oak, 4.0 to 3.0 for poplar, and 2.3 to 1.4 for hornbeam.

In a 9-month-long test [41], 20 × 20 × 300 mm samples of oak and black locust and 10 × 10 × 150 mm samples of teak wood were tested to detect changes in physical properties. After 9 months, the density was reduced to 83%, 75%, and 97% for black locust, oak, and teak, respectively. Mass was 91%, 82%, and 99% of the initial mass of black locust, oak, and teak, respectively.

In a 2-year-long test [42], five stakes of beeswax-impregnated beech and Scots pine with dimensions of 25 × 50 × 500 mm were tested. After 1 year of exposure, the decay rating due to beeswax impregnation decreased from 3.6 to 1.6 for beech and from 3.2 to 0.6 for Scots pine. After 2 years of exposure, it was from 4.0 to 2.2 for beech and 3.2 to 1.4 for Scots pine.

### 3.3. Rate of Degradation

Mold started growing and cracks appeared on untreated stakes already after 1 month. There was no sign of decay on the acetylated hornbeam after 1 year. Wasp stripping was observed on untreated hornbeam and beech after 8 months. Moss appeared on Scots pine stakes after 10 months. Signs of fungal decay were visible on untreated stakes already after 5 months. After 1 year, three untreated hornbeam and one untreated beech stake broke during evaluation. Signs of soft rot, white rot, and insect damage were observed. Whitish discoloration, long-fibered splinter and fibered structure indicated white rot attack. Mushrooms of *Coprinellus micaceus* were identified on untreated hornbeam stakes. Soft rot decay attacks hardwoods more than softwoods because the lignin is more methoxylated in hardwoods [1,12].

After 15 months, mushrooms of *Bjerkandera adusta* and *Coprinellus micaceus* formed on beech stakes. Moss was found later on acetylated hornbeam stakes after 17 months. One of the acetylated hornbeam stakes showed signs of brown rot decay after 18 months (Figure 3). The rate of white rot and soft rot decay, as well as insect damage in untreated stakes, worsened after another year.

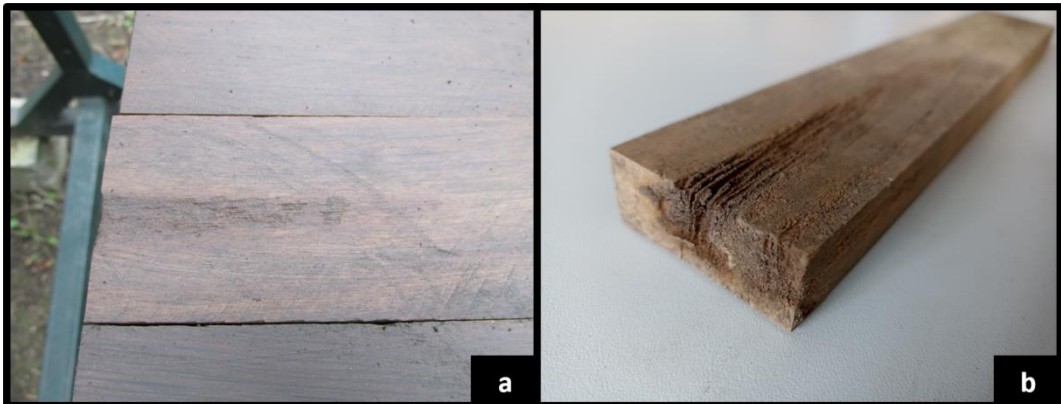

**Figure 3.** Brown rot in acetylated hornbeam stake Number 7 after 1.5 years (**a**) and 5.5 years (**b**).

After 2 years of exposure, three hornbeam and seven beech stakes failed the test. Scots pine stakes were heavily degraded by soft rot and brown rot. All untreated hornbeam stakes failed after 3.5 years, while beech stakes failed already after 2.5 years. Scots pine stakes started to fail after 3.5 years, and all of them broke after 6 years. Small cubic breaks, softened surface, dark discoloration, insect damage, and moss were observed on them. The soil was probably less optimal for the growth of brown rot fungi, which can explain the slower decaying mechanism compared to soft rot and white rot.

After 5.5 years, acetylated hornbeam stake Number 7 was taken out from the test to examine its properties as the only locally decayed stake. The local decayed part of the sample may have been a "wet pocket" in the untreated wood, which was acetylated at a lower degree due to the higher moisture content. If it had a lower degree of acetylation, it could have been more prone to be attacked by fungi. White rot has the same preference in case of acetylated wood as untreated: it degrades acetylated hardwoods faster than acetylated softwoods [11]. On the other hand, acetylated wood is less susceptible to white rot than brown rot, as the decaying mechanism of white rot depends on the presence of hydroxyl groups in the cell walls, which help initialize the attack by hydrolyzing enzymes. The higher the WPG, the more hydroxyl groups in lignin are exchanged by acetyl groups during acetylation, and the less likelihood that white rot fungi can diffuse into the cell walls and attack. Brown rot was observed on acetylated wood, as it can attack it even at this WPG (15%) or higher due to its non-enzymatic system [11,18].

The test was valid according to EN 252: 2016 because at least 75% of the untreated stakes were rated 4, and the acetylated stakes were rated not less than 2. In addition, local signs of decay existed on at least one acetylated stake.

Figure 4 summarizes the rating and lifetime of each stake. The average lifetime of untreated hornbeam, beech and Scots pine was 2.0 ± 1.0, 1.8 ± 0.6, and 5.1 ± 1.0 years, respectively., and all of the stakes broke. Acetylated hornbeam stake number 7 had mild local brown rot decay, while the others showed no signs.

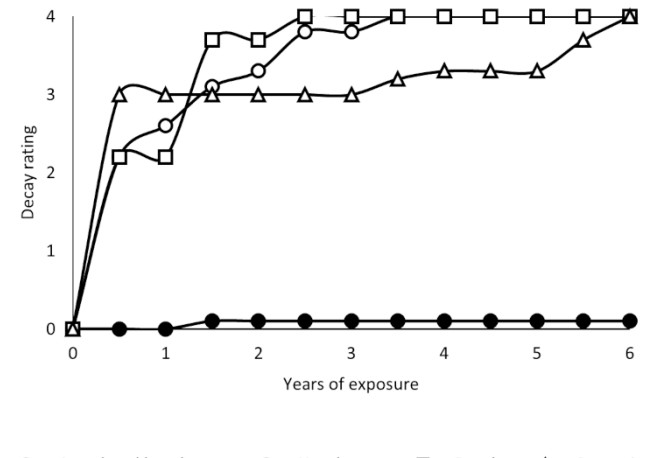

**Figure 4.** Average rating of in-ground stakes during exposure period (2016–2022). Decay rating or durability class is given according to EN 252 (2015): 0—sound, 1—slight attack, 2—moderate attack, 3—severe attack, 4—failure.

On average, one-third of mass and one-third of density were lost during the exposure in the case of untreated stakes (Table 3). The local brown rot-decayed acetylated hornbeam sample lost 6% of its mass and density after 5.5 years of field test. Depth of degradation was calculated by halving the thickness loss of samples, which was 0.02, 0.18 ± 0.34, 0.34 ± 0.11, and 1.98 ± 0.82 mm for acetylated hornbeam, beech, hornbeam, and Scots pine sapwood, respectively. In Scots pine, there was a great difference between the degradation of earlywood and latewood because earlywood was more susceptible to fungal attack. There were no correlations found between density, annual ring density, and lifetime.

**Table 3.** Summary of results of field soil trial: dry mass and dry density loss of tested wood species after 6 years. Average values are presented with standard deviation in brackets.

| Wood Species | Dry Mass Loss (%) | Dry Density Loss (%) |
|---|---|---|
| Acetylated hornbeam * | 6 (0) | 6 (0) |
| Hornbeam | 28 (12) | 28 (12) |
| Beech | 30 (6) | 29 (6) |
| Scots pine sapwood | 39 (9) | 23 (10) |

\* no failure occurred, but it was taken from the test for examination.

### 3.4. Microscopic Evaluation

After visual inspection, we inspected acetylated hornbeam stake Number 7 with a microscope to investigate the biological damage. The results show that some parts were not damaged or only the presence of hyphae was observed without cell wall damage. This corresponds to other reports in the literature [11,12,18,19,30]. Hyphae can colonize cell lumina and ray cells of acetylated wood because they penetrate the wood across open ways like vessel lumina and rays, then into fiber cell lumina through inter-fiber pits and cross-fields between rays and fibers [11,12].

In the case of decayed parts in acetylated hornbeam, typical signs of soft and brown rot decay have been found, such as hyphae and cavity formation in cell walls, amorphous cell walls, erosion, and the thinning of cell walls with intact middle lamella (Figure 5) [1,3].

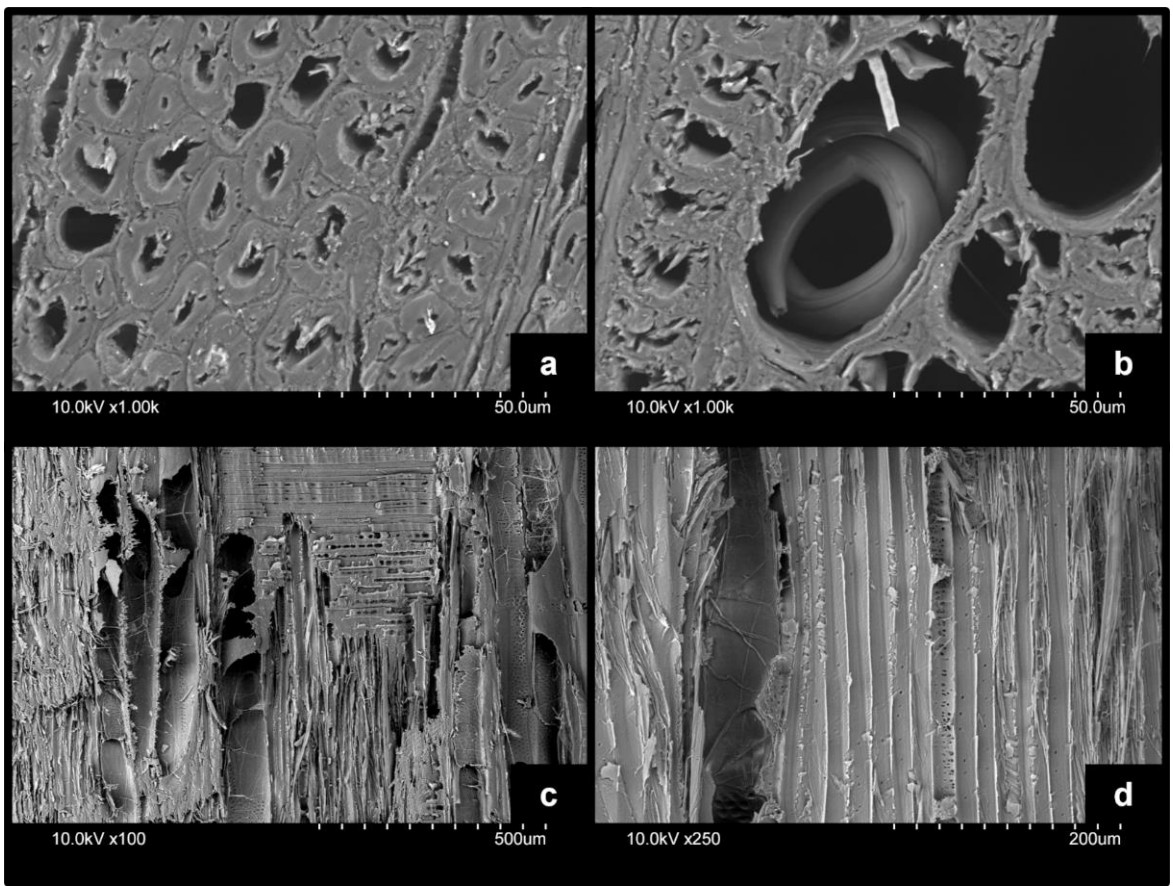

**Figure 5.** SEM pictures of cross section (above) and longitudinal section (below) of partly decayed acetylated hornbeam stake: (**a**) non-decayed parts with thick cell walls, (**b**) worm in cell lumen, (**c**) disappeared bordered pits, fibrous tissue, (**d**) hyphae in lumen, gradual thinning of cell walls, typical signs of soft rot decay.

In untreated stakes, the microscopic investigation revealed the rate of damage by soft rot and white rot. Fungal decay caused massive damage with cell wall thinning, the gradual disappearance of rays and bordered pits, erosion, fibrous tissue, and disappeared middle lamella (Figures 6–8). Large boreholes, large round holes in the cell wall, and hyphae with clamp connections could also be found in the microscopic sections [1,3].

### 3.5. Fourier Transform Infrared Spectroscopy

The wood that originated from the "initial decay" area was relatively brittle, which made working with the mortar and pestle relatively easy. The piece of wood that originated from the intact part of the board was much less brittle and required more force than could be applied through the mortar and pestle.

Figure 9 displays typical spectrums for wood from the two different parts. The dotted line corresponds to the spectrum of acetylated hornbeam wood part with initial decay, while the solid line represents the spectrum of acetylated hornbeam wood that is considered normal in appearance. The spectra have been shifted with respect to each other for better visualization.

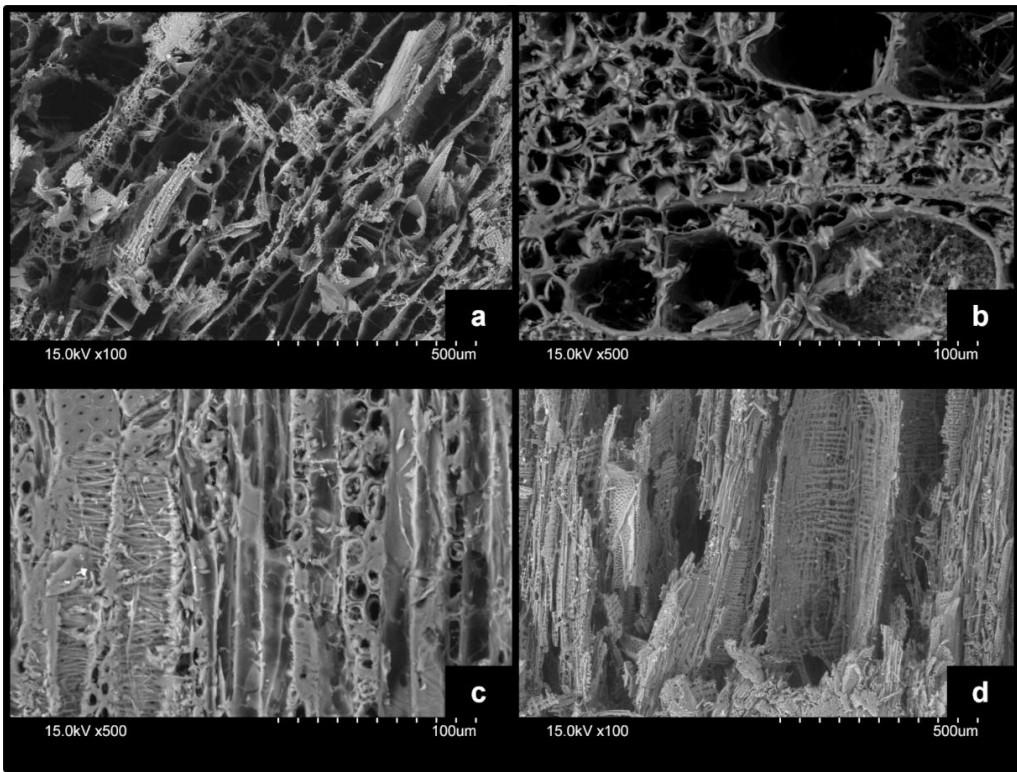

**Figure 6.** SEM pictures of cross section (above) and longitudinal section (below) of locally decayed hornbeam wood: (**a**) cell wall thinning, (**b**) fungal hyphae in lumen, (**c**) gradual disappearance of bordered pits, (**d**) fibrous tissue.

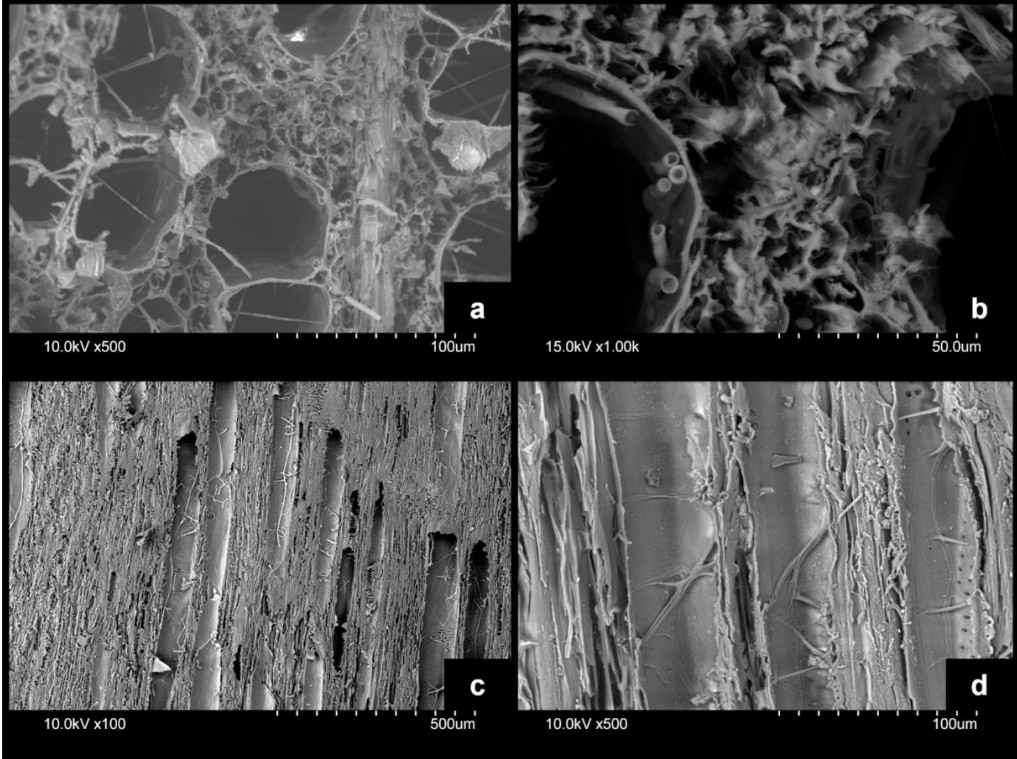

**Figure 7.** SEM pictures of cross section (above) and longitudinal section (below) of decayed beech wood: (**a**) fibrous tissue, (**b**) hyphae in lumen, (**c**) damaged bordered pits, (**d**) growing hyphae along vessels.

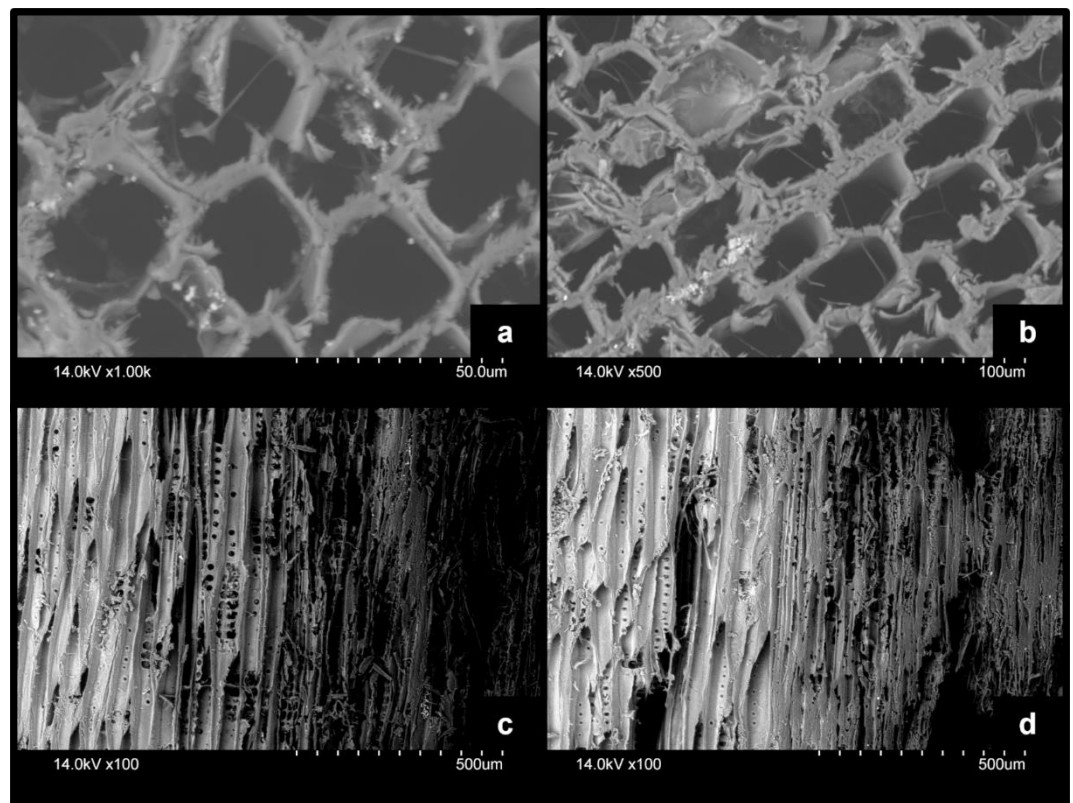

**Figure 8.** SEM pictures of cross section (above) and longitudinal section (below) of decayed Scots pine wood: (**a**) thinned cell wall structure, (**b**) fungal hyphae in lumina, (**c**) fibrous structure, (**d**) gradual disappearance of bordered pits on decay border.

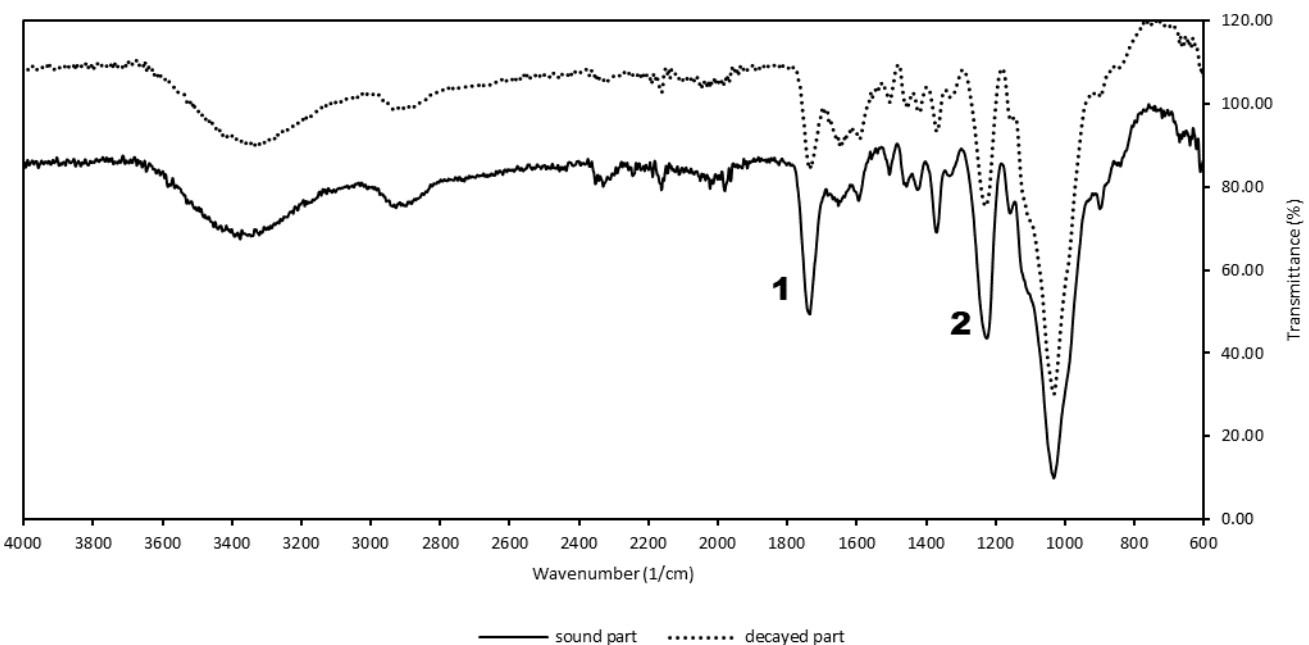

**Figure 9.** FTIR spectra of acetylated hornbeam wood powder with initial decay (dotted line) and no visible decay (solid line).

The largest peak with maximum intensity and absorbance was caused by C-O vibration in hemicellulose (1034 cm$^{-1}$ for decayed, and 1032 cm$^{-1}$ for sound part) [43]. This peak was considered constant (absorbance equals 1) in order to compare the 1730 cm$^{-1}$ and

1230 cm$^{-1}$ peaks between both spectra. Thus, the PHR between examined peak and 1030 cm$^{-1}$ equaled the absorbance of the examined peak.

The intensity at peak (1) was relatively smaller for the decayed part (1738 cm$^{-1}$, %T 64.41, %A 0.191) compared to the non-decayed part (1734 cm$^{-1}$, %T 49.45, %A 0.306). This peak was caused by unconjugated C=O (carbonyl) bond stretching in acetyl in hemicelluloses [7,43].

There was also a lower absorption at peak (2), which was caused by C-O stretching in the acetyl groups in hemicellulose xylan and mannosan [43]. The amount of acetyl-groups present in the decayed part of the wood (1228 cm$^{-1}$, %T 55.25, %A 0.258) was lower than in the rest of the wood (1224 cm$^{-1}$, %T 43.62, %A 0.360).

This means that the "affected part" had a much lower acetyl content. It was likely a wet pocket or a place with lower acetylation. This stake had the lowest annual ring density (7 per 2 cm) and the lowest WPG (13.55%) among other acetylated boards from which the stakes were taken.

In a research study concerning acetylated Radiata pine exposed to brown rot fungi [44,45], fungal deterioration was enabled by a de-acetylation mechanism during an initial lag phase. It was concluded that the bonds between chemical groups and biopolymers can be attacked and broken by fungi in optimal conditions for decay. This could also explain partially the lower acetyl content of the acetylated hornbeam stake, although it was of a different wood species. This also means that the acetyl content may not have been as low initially, as it was measured by FTIR.

## 4. Conclusions

To date, long-term field tests have shown that acetylated hornbeam exhibits greater resistance against fungal decay, mold, insects, and moisture than untreated hornbeam, beech, and Scots pine sapwood do. These tests were evaluated every 6 months.

The testing field had a sandy loam type of soil, being pH neutral, rich in nutrients, calcium carbonate, humus, and having water management properties favorable for fungi. The wet period was typical for the whole year, and warm periods were more than half a year long, which also enabled fungal growth.

After 6 years of exposure, there was no sign of decay on almost all acetylated hornbeam stakes, which became dry shortly after being taken from the soil. Acetylated hornbeam stake number 7 had local signs of superficial brown rot decay after 1.5 years, which slowly worsened over the years. FTIR analysis indicated the affected area of this stake had a much lower acetyl content compared to sound areas of the stake. The local lower acetyl content could have been caused by a wet pocket in the untreated hornbeam.

On the untreated hornbeam stakes, insect damage, soft rot decay, white rot decay, wasp stripping, and cracks were observed during the test. There was one stake that already broke after a half-year of exposure, but all hornbeam stakes eventually broke after 3.5 years. On average, the lifetime of hornbeam stakes was 2 years, which means that acetylation lengthened its lifetime by at least three times (in soil and climate of testing field).

Beech had the same types of decay and damage as hornbeam did. There was one stake that broke after only a half year of exposure, but all beech stakes broke eventually after 2.5 years. On average, the lifetime of beech stakes was 1.8 years.

Stakes made from Scots pine sapwood had insect damage, soft rot decay, white rot decay, brown rot decay, moss, and cracks. Although the signs of decay were severe, there were only two stakes that broke after 3.5 years, and the last stakes lasted in the soil for 6 years. On average, the lifetime of Scots pine stakes was 5.1 years. The soil and the climate were probably more favorable for soft rot and white rot decay, which attack pine less rapidly than hardwoods such as beech and hornbeam.

Based on these findings, acetylated hornbeam shows promising results for further research, and for the production of exterior products such as furniture, fencing, decking, cladding, paneling, playground elements, etc. Instead of burning hornbeam wood right away, acetylation can widen its fields of use in order to lengthen its lifespan.

In future work, acetylated hornbeam stakes having different WPG levels should be tested to evaluate the threshold value of maximum durability. The durability of other underutilized wood species (e.g., Turkey oak) should be assessed after acetylation as well.

**Supplementary Materials:** The following supporting information can be downloaded at: https://www.mdpi.com/article/10.3390/f13071003/s1, Table S1: Average monthly breakdown of weather parameters; Table S2: Annual breakdown of weather parameters.

**Author Contributions:** Conceptualization, F.F.; methodology, F.F.; validation, F.F., A.B., M.B. and R.N.; formal analysis, F.F.; investigation, F.F., M.B. and B.B.-V.; resources, F.F., M.B. and A.B.; writing—original draft preparation, F.F. and A.B.; writing—review and editing, M.B. and R.N.; visualization, F.F.; supervision, R.N.; project administration, R.N.; funding acquisition, R.N. All authors have read and agreed to the published version of the manuscript.

**Funding:** This article was made in frame of the project TKP2021-NKTA-43, which has been implemented with the support provided by the Ministry of Innovation and Technology of Hungary (successor: Ministry of Culture and Innovation of Hungary) from the National Research, Development and Innovation Fund, financed under the TKP2021-NKTA funding scheme.

**Data Availability Statement:** Not applicable.

**Acknowledgments:** The authors would like to thank Pál Balázs, Márton Kiss, and Róbert Roszik for weather data, and Norbert Horváth for his useful insights and help in the preparatory work. Accsys Technologies is acknowledged for performing the acetylation treatment.

**Conflicts of Interest:** The authors declare no conflict of interest.

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
