# Peer review of "Biological Durability of Acetylated Hornbeam Wood with Soil Contact in Hungary"

_forests, doi:10.3390/f13071003_

Round 1

Reviewer 1 Report

The topic of the research work and manuscript is really interesting, well prepared and provides new information. However there are some issues to be addressed towards its quality improvement.  In line 13, I would rather say "focus on/emphasized on" instead of "concentrated on". Probably the word "pole" seems more suitable instead of "stakes". In lines 20-21, the sentence "revealed that the stake had lower acetyl content" should be clarified (which are exactly these stakes). In lines 35-45, please provide reference. In lines 51-53, please provide reference. In lines 55-56, do you mean that the methods of measuring the strength toughness and impact bending strength are the most precise methods of recording the early stages of decay? (please clarify). In the frame of the manuscript quality improvement, I would propose to provide the level of protection improvement of acetylated wood species (line 68). In line 72, I'd rather use "moisture content" instead of humidity. In lines 75-85, you should provide references (I would propose to use DOI: 10.3390/f10121111 in this point to support your statements). In line 84, I would replace "more moisture" to "higher". In line 100, do you probably mean "against" (instead of "compared to"). In 100-104, 122-124, please provide reference. Even though the introduction chapter describes well the state-of-the-art, it seems to be too extended/a bit difficult to be read, it needs to be shortened keeing and highlighting the most significant findings of literature. For exmaple 108-112 lines have been already referred earlier in the introction theoretical approach. You rather use "hemicellulose" in plural form. Lines 134-136 have already been refered earlier. Lines 143-146: the meaning is not very clear. In 147-148 lines, you could add a comment for the insects that are not affected by moisture content (contrary to fungi). In line 172, add the word "species". The identification of literature gap has not been properly implemented in my opinion and improvement changes should be conducted in this direction. Add a brief comment on why you have chosen hornbeam to be exmined in this paper. In lines 199-201, do you believe that your findings could be comparable to the liteaure findings given the modifications you conducted to the standard process. In table 1, the beech material used was sapwood/heartwood/both? In materials and methods chapter, the part of "2.6. Fourier Transform Infrared Spectroscopy (FTIR)" needs clarification, why did you use only this specimen? why did not you use the control specimens as well. In line 372, refer the number and year of reference as well in the first time mentioned in the sentence (not in the end). You should describe if any statistical analysis was performed in the results or otherwise provide at least standard deviation values in tables and graphs. The figures provided by the authors are very useful to the readers. In line 491, "were evaluated". I would recommend also to improve/rephrase conclusions keeping/repeating only the most significant findings and drawing some conclusions from your findings (where the materials could be applied, what could be improved, future research on acetylation field or these species etc.) to highlight also the significance of this work. 

Reviewer 2 Report

The manuscript concerns field test of the decay resistance of acetylation hornbeam compared with untreated wood samples of a variety of species. In general, the manuscript contains lots and lots and lots of information and text that appear irrelevant (or of minor relevance) to the present study. For instance, the introduction is hard to follow, because the purpose of the study is only explained very late. So, the introduction now reads as a long list of information about decay of wood, but the relevance to this particular study is not easy to grasp for the reader.

Therefore, the authors need to heavily revise the manuscript by shortening it substantially (by approximately 60-80 %) to only contain information relevant for this study, and by clearly explaining upfront the motivation of this study, i.e. why hornbeam? Why acetylation? Lines 66-70 are good and could be used in this argumentation. However, they are immediately followed by a section (lines 71-94) reviewing all sorts of information unrelated to acetylation.

Below is a list of specific comments to locations in the text:

1) L108-112: “The reasons that wood modification is effective against fungal decay are the following: the equilibrium moisture content is lowered in modified wood, hence it is harder for fungi to obtain the moisture required for decay, and there is physical blocking of the entrance of decay fungi to micropores of the cell walls, and/or inhibition of the action of specific enzymes [16, 17].” This is an  example of information overload about possible reasons for decay resistance in acetylated wood. The micropore blocking mechanism was a hypothesis that later turned out not be be supported by experimental results, similar to the imhibition/non-recognition theory. Thus, the authors need to substantially shorten the introduction, leaving only the part of state-of-the-art relevant for the present study.

2) L172-181: “Hornbeam (Carpinus betulus L.) is a diffuse-porous wood; it does not form colored heartwood and has low natural durability (DC 5 according to EN 350: 2016). Hence, it is not recommended for outdoor use. Without soil contact and under a roof, it can last 35 years, but with soil contact, its wood will decompose due to fungal decay after only two to three years. Hornbeam also tends to spalt, which is a major consideration for lumber production. It is highly resistant to strong acids and alkalis. It is chemically inactive and has a small amount of wood extractives. During weathering, it becomes grey in a short time and surface checks become apparent. The most frequently used and most durable species for outdoor applications in Hungary are oak, black locust, Scots pine, and larch. Spruce is also common, but it needs to be treated because of its low durability [26, 27].” Why is the information about resistance to acids and alkalis relevant? What about the concentration of extractives? There is so much information in just this section that seems completely irrelevant to the present study – and similarly with the other parts of the introduction.

3) L199-207: There is no information about how the hornbeam specimens were acetylated and their WPG.

4) L295-322: What is the purpose of the soil testing? How is it relevant for this particular study given that all specimens are exposed in the same soil? It is of course fine to characterize the exposure conditions but a lot of text is spent on describing the soil results without much reflection of the relevance for the reader. Could this information be moved to a Supplementary Material?

5) L324-335: This paragraph is also much longer than necessary. Please heavily condense it.

6) L336-337: “The Hungarian field is probably faster in terms of decay compared to hot and dry territories.” Is this a speculation relevant for understanding the results of this study?

7) Tables 3 and 4 – how relevant are these numbers? Could they be removed or moved to Supplementary Material?

8) L386-388: “There are hot spots in Europe with relatively high decay potential (maximum 81) on the west coast of Norway and in Ireland, the UK, and France. Generally, the decay potential decreases towards the east due to the increasingly continental character of the climate.” Is this a relevant discussion for this particular study?

9) L414-417: “The local decayed part of the sample may have been a ‘wet pocket’ in the untreated wood, which was acetylated at a lower degree due to the higher moisture content. Because it had a lower degree of acetylation, it was more prone to be attacked by fungi.” These two sentences are confusing. The first sentence indicate a hypothesis concerning a lower degree of acetylation, whereas the second sentence appears to take the lower degree of acetylation as a fact. Was there any examination done or is it just a hypothesis?

10) L471-473: “The red spectrum is from the “initial decay” wood, while the blue spectrum is measured from wood that is considered normal in appearance.” There are no blue and red spectra in Figure 9, only black lines.

11) L478-479: “The peak at 1740 cm-1 is relatively smaller for the decayed part compared to the non-decayed part.” This is very difficult for the reader to see. Typically, IR spectra are shown with absorbance on the y-axis. Were the spectra baseline corrected and normalized? Otherwise, it might be difficult to interpret too much from the data.

12) L484-485: “The conclusion is that the “affected part” has a much lower acetyl content. It was likely a wet pocket or a place with lower acetylation.” The lower degree of acetylation could also partially be explained by deacetylation by the fungi as shown by Beck et al. (https://doi.org/10.1016/j.ibiod.2018.09.009, https://doi.org/10.1016/j.ibiod.2021.105257). Thus, it is not given that the acetyl content initially was as low as measured on the decayed part of the sample.

Round 2

Reviewer 1 Report

As I have checked the authors have implemented the proposed changes in the revised verion of manuscript towards the improvement of their work. Almost all the changes have been implemented and in my opinion, the manuscript is well-prepared and organized.

Author Response

Thank you very much!

Reviewer 2 Report

The authors have followed my suggestion of condensing the manuscript which has markedly improved it. However, Lines145-169 still contain a lot of information that does not appear very relevant for the present study. Could this perhaps also be moved or substantially shortened?

Furthermore, the authors need to properly analyse the FTIR-data in order to discuss and interpret the results:

L512-523: The FTIR method is not properly explained or the data have not been properly analysed. It is not enough just to perform the measurements and then compare the results without baseline correction or normalization. This is absolutely necessary in order to interpret the results of Figure 9.

L798-811: It is not possible to discuss or interpret the results without proper analysis of the experimental data. The spectra should be baseline corrected and normalized before peaks heights of the different materials are compared.
